# Trajectory Convolution for Action Recognition

**Yue Zhao**
Department of Information Engineering
The Chinese University of Hong Kong
zy317@ie.cuhk.edu.hk

**Yuanjun Xiong**
Amazon Rekognition
yuanjx@amazon.com

**Dahua Lin**
Department of Information Engineering
The Chinese University of Hong Kong
dhlin@ie.cuhk.edu.hk

## Abstract

How to leverage the temporal dimension is one major question in video analysis. Recent works [47, 36] suggest an efficient approach to video feature learning, *i.e.*, factorizing 3D convolutions into separate components respectively for spatial and temporal convolutions. The temporal convolution, however, comes with an implicit assumption – the feature maps across time steps are well aligned so that the features at the same locations can be aggregated. This assumption can be overly strong in practical applications, especially in action recognition where the motion serves as a crucial cue. In this work, we propose a new CNN architecture *TrajectoryNet*, which incorporates *trajectory convolution*, a new operation for integrating features along the temporal dimension, to replace the existing temporal convolution. This operation explicitly takes into account the changes in contents caused by deformation or motion, allowing the visual features to be aggregated along the the motion paths, *trajectories*. On two large-scale action recognition datasets, Something-Something V1 and Kinetics, the proposed network architecture achieves notable improvement over strong baselines.

## 1 Introduction

The past decade has witnessed significant progress in action recognition [37, 38, 29, 42, 1], especially due to the advances in deep learning. Deep learning based methods for action recognition mostly fall into two categories, two-stream architectures [29] with 2D convolutional networks and 3D convolutional networks [34]. Particularly, the latter has demonstrated great potential on large-scale video datasets [19, 25], with the use of new training strategies like transferring weights from pretrained 2D CNNs [42, 1].

However, for 3D convolution, several key questions remain to be answered: (1) 3D convolution involves substantially increased computing cost. *Is it really necessary?* (2) 3D convolution treats the spatial and temporal dimensions uniformly. *Is it the most effective way for video modeling?* We are not the first to raise such questions. In recent works, there have been attempts to move beyond 3D convolution and further improve the efficiency and effectiveness of joint spatio-temporal analysis. For instance, both *Separable-3D (S3D)* [47] and *R(2+1)D* [36] obtain superior performance by factorizing the 3D convolutional filter into separate spatial and temporal operations. However, both methods are based on an *implicit* assumption that the feature maps across frames are well aligned so that the features at the same locations (across consecutive frames) can be aggregated via temporal convolution. This assumption ignores the *motion* of people or objects, a key aspect in video analysis.

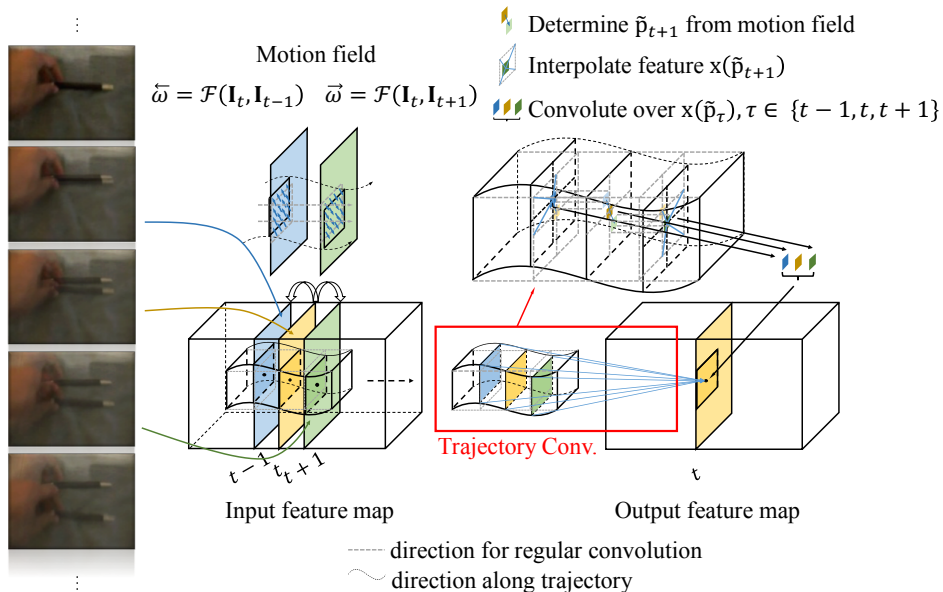

Motion field
$\overleftarrow{\omega} = \mathcal{F}(\mathbf{I}_t, \mathbf{I}_{t-1})$  $\overrightarrow{\omega} = \mathcal{F}(\mathbf{I}_t, \mathbf{I}_{t+1})$

Determine $\tilde{\mathrm{p}}_{t+1}$ from motion field
Interpolate feature $\mathrm{x}(\tilde{\mathrm{p}}_{t+1})$
Convolute over $\mathrm{x}(\tilde{\mathrm{p}}_\tau), \tau \in \{t-1, t, t+1\}$

Trajectory Conv.

$t-1$  $t$  $t+1$

Input feature map

Output feature map

$t$

- - - - direction for regular convolution
~~~ direction along trajectory

Figure 1: Illustration of our trajectory convolution. Given a sequence of video frames (`left`) and its corresponding input feature map of size $C \times T \times H \times W$ (`bottom-middle`; the dimension of channels $C$ is simplified as one for clarity), in order to calculate the response of a specific point at time step $t$, we leverage the motion fields $\overleftarrow{\omega}$ and $\overrightarrow{\omega}$ (`top-middle`; the arrows in blue denote the motion velocity) to determine the sampling location at neighboring time step $t-1$ and $t+1$ in the sense of tracking along the motion path. The response is denoted on the output feature map (`bottom-right`). The operation of trajectory convolution (denoted in a red box) is illustrated on the `top-right`. This figure is best viewed in color.

A natural idea to address this issue is to track the objects of interest and extract the features along their motion paths, *i.e.*, trajectories. This idea has been explored in previous works [33, 37, 38, 41]. The most recent work along this direction is the *Trajectory-pooled Deep-convolutional Descriptor (TDD)* [41], which aggregates off-the-shelf deep features along trajectories. However, in this method, the visual features are derived *separately* from an existing deep network, just as a replacement of hand-crafted features. Hence, a question emerges: *can we learn better video features in conjunction with feature tracking?*

In pursuit of this question, we develop a new CNN architecture for learning video features, called *TrajectoryNet*. Inspired by the Separable-3D network [36, 47], our design involves a cascade of convolutional operations respectively along the spatial and temporal dimensions. A distinguishing feature of this architecture is that it introduces a new operation, namely the *trajectory convolution*, to take the place of the standard temporal convolution. As shown in Figure 1, the *trajectory convolution* operates along the trajectories that trace the pixels corresponding to the same physical points, rather than at fixed pixel locations. The trajectories can be derived from either a precomputed optical flow field or a dense flow prediction network trained jointly with the features. The standard temporal convolution can be seen as a special case of the trajectory convolution where all pixels are considered to be stationary over time.

Experimental results on Something-Something V1 and Kinetics datasets show that by explicitly taking into account the motion dynamics in the temporal operation, the proposed network obtains considerable improvements over the Separable-3D, a competitive baseline.

## 2   Related Work

**Trajectory-based Methods for Action Recognition**   Action recognition in videos has been greatly advanced thanks to the up-springing of powerful features. It was firstly tackled by extracting spatial-temporal local descriptors [39] from space-time interest points [20, 46]. These successful local features include: Histogram of Oriented Gradients (HOG) [3], Histogram of Optical Flow (HOF) [21], and Motion Boundary Histogram (MBH) [4].

Over the years, it was recognized that the 2D space domain and 1D time domain have different characteristics and should be handled in a different manner intuitively. As of the motion modeling in the temporal domain, trajectories have been a powerful intermediary to convey such motion information. Messing *et al* [24] used a KLT tracker [22] to extract feature trajectories and applied log-polar uniform quantization. Sun *et al* [33] extracted trajectories by matching SIFT feature between frames. These trajectories are based on sparse interest points, which have been later proved to be inferior to dense sampling. In [37], Wang *et al* used dense trajectories to extract low-level features within aligned 3D volumes. An improved version [38] increased recognition accuracy by estimating and compensating the effect of camera motion. [37, 38] also revealed that trajectory itself can serve as a component of descriptors in the form of concatenated displacement vectors, which was consolidated by deep learning methods [29].

Wang *et al* first proposed TDD in [41] to introduce deep features to trajectory analysis. It conducts trajectory-constrained pooling to aggregate deep features into video descriptors. However, the backbone two-stream CNN [29], from which the deep feature is extracted, is learned from very short frame snippets and is unaware of the information of temporal evolution. In addition, all of these trajectory-aligned methods rely on encoding methods such as Fisher vectors (FV) [45] and vectors of locally aggregated descriptors (VLAD) [14] and an extra SVM is needed for classification, which prohibits end-to-end training. To sum up the discussion above, we provide a comparison of our approach with previous works on action recognition in Table 1.

Table 1: A comparison of our approach with existing methods.

| Method | Use deep feature? | Feature tracking? | End-to-end? |
|---|---|---|---|
| STIP [20] | ✗ | ✗ | ✗ |
| DT [37], iDT [38] | ✗ | ✓ | ✗ |
| TSN [42], I3D [1] | ✓ | ✗ | ✓ |
| TDD [41] | ✓ | ✓ | ✗ |
| TrajectoryNet (Ours) | ✓ | ✓ | ✓ |

**Action Recognition in the Context of Deep Learning** Deep convolutional neural networks based models have been widely applied to action recognition [18, 29, 34], which can be mostly categorized into two families, *i.e.* two-stream networks [29] and 3D convolution networks [15, 34]. Recently, 3D convolutonal network has drawn attention since Carreira *et al* introduced Inflated-3D models [1] by inflating an existing 2D convolutional network to its 3D variant and training on a very large action recognition dataset [19]. Tran *et al* argued in [36] that factorizing 3D convolutions into separable spatial and temporal convolutions obtains higher accuracy. Similar phenomenon is also observed in Separable-3D models by Xie *et al* [47]. Wang *et al* incorporated multiplicative interaction into 3D convolution for modeling relation in [40]. All of these modifications are focused on the single modality, *i.e.* the appearance branch.

Apart from network architectural designs, another direction is to exploit the interaction of appearance and motion information of action. Feichtenhofer *et al* explored the strategies of spatio-temporal fusion of two-stream networks at earlier stages in [7]. Such attempts are mostly simple manipulation of feature such as stacking, addition [7], and multiplicative gating [6].

**Motion Representation using Convolutional Networks** Optical flow has been used as a generic representation of motion as well as trajectory in particular for decades. As a competitive counterpart to the classical variational approaches [10, 31], many parametric models based on CNN have been recently proposed and achieved promising results in estimating optical flow. These include, but are not limited to, the FlowNet family [5, 11], SpyNet [26], and PWC-Net [32]. The aforementioned models are learned in a supervised manner on large-scale simulated flow datasets [5, 23], possibly leaving a large gap between simulated animations and real-world videos. Also, these datasets are designed for accurate flow prediction, which is possibly not appropriate for motion estimation of human action due to inhomogeneity of displacement across optical flow dataset and human action dataset, as revealed in [11]. As of the network architecture, most models require parameters in the magnitude of $10^7 \sim 10^8$, which both prohibits being plugged into action recognition networks as a submodule and causes too much computational cost. Zhu *et al* proposed MotionNet [51] to learn dense flow fields in an unsupervised manner and plugged it into a two-stream network [29] to be finetuned for action recognition task. The MotionNet is relatively light-weighted and can accept a sequence of multiple images. However, this is only used to substitute the pre-calculated optical

flow while maintaining the conventional two-stream architecture. Zhao *et al* proposed an alternative representation based on cost volume for efficiency at the cost of degraded quality of motion field [49].

**Transformation-Sensitive Convolutional Networks**   Conventional CNN operates on fixed locations in a regular grid, which limits its ability to modeling unknown geometric transformations. Spatial Transform Networks (STN) [13] is the first to introduce spatial transformation learning into deep models. It estimates a global parametric transformation on which the ordinary feature map is warped. Such warping is computationally expensive and the transformation is considered to be universal across the whole image, which is usually not the case for action recognition, since different body parts have their own movement. In Dynamic Filter Networks [16], Xu *et al* introduce dynamic filters which are conditioned on the input and can change over samples. This enables learning local spatial transformations. Deformable Convolutional Network (DCN) [2] achieves similar local transformation in a different way. While maintaining filter weights invariant to the input, the proposed deformable convolution first learns a dense offset map from the input, and then applies it to the regular feature map for re-sampling. The proposed trajectory convolution is inspired by the deformable sampling in DCN and utilizes it for feature tracking in the spatiotemporal convolution operations.

## 3   Methods

The TrajectoryNet model is built with the trajectory convolution operation. In this section, we first introduce the concept of trajectory convolution. Then we illustrate the architecture of TrajectoryNet. Finally we describe the approach to learning the trajectory together with the trajectory convolution.

### 3.1   Trajectory Convolution

In the context of separable spatial temporal 3D convolution, the 1D temporal convolution is conducted pixel-wise on the 2D spatial feature map along the temporal dimension. Given input feature maps $\mathbf{x}_t(\mathbf{p})$ at the $t$-th time step, the output feature $\mathbf{y}_t(\mathbf{p})$, at position $\mathbf{p} = (h, w) \in [0, H] \times [0, W]$, is calculated by the inner product of input feature sequences at same spatial position across neighboring frames and the 1D convolution kernels.

By revisiting the idea of trajectory modeling in the action recognition literature, we introduce the concept of *trajectory convolution*. In trajectory convolution, the convolutional operation is done across irregular grids such that the sampled positions at different times correspond to the same physical point of a moving object. Formally, parameterized by the filter weight $\{\mathbf{w}_\tau : \tau \in [-\Delta t, \Delta t]\}$ with kernel size $(2\Delta t + 1)$, the output feature $\mathbf{y}_t(\mathbf{p})$ is calculated as

$$\mathbf{y}_t(\mathbf{p}) = \sum_{\tau=-\Delta t}^{\Delta t} \mathbf{w}_\tau \mathbf{x}_{t+\tau}(\widetilde{\mathbf{p}}_{t+\tau}). \tag{1}$$

Following the formulation of trajectory in [37], the point $\mathbf{p}_t$ at frame $t$ can be tracked to position $\widetilde{\mathbf{p}}_{t+1}$ at next frame $(t+1)$ in the presence of a forward dense optical flow field $\overrightarrow{\omega} = (u_t, v_t) = \mathcal{F}(\mathbf{I}_t, \mathbf{I}_{t+1})$ using the following equation

$$\widetilde{\mathbf{p}}_{t+1} = (h_{t+1}, w_{t+1}) = \mathbf{p}_t + \overrightarrow{\omega}(\mathbf{p}_t) = (h_t, w_t) + \overrightarrow{\omega}|_{(h_t, w_t)}. \tag{2}$$

For $\tau > 1$, the sample position $\widetilde{\mathbf{p}}_{t+\tau}$ can be calculated by applying Eq. (2) iteratively. To track to the previous frame $(t-1)$, a backward dense optical flow field $\overleftarrow{\omega} = (u_t, v_t) = \mathcal{F}(\mathbf{I}_t, \mathbf{I}_{t-1})$ is used likewise.

Since the optical flow field is typically real-valued, the sampling position $\widetilde{\mathbf{p}}_{t+\tau}$ becomes fractional. Therefore, the corresponding feature $\mathbf{x}(\widetilde{\mathbf{p}}_{t+\tau})$ is derived via interpolation with a specific sampling kernel $G$, written as

$$\mathbf{x}(\widetilde{\mathbf{p}}_{t+\tau}) = \sum_{\mathbf{p}'} G(\mathbf{p}', \widetilde{\mathbf{p}}_{t+\tau}) \cdot \mathbf{x}(\mathbf{p}'). \tag{3}$$

In this paper, we will not go deeper into the usage of different choices of sampling kernels $G$ and use the bilinear interpolation as default.

## 3.2 Relation with Deformable Convolution

The original deformable convolution is introduced for 2D convolution. But it is natural to extend it to the 3D scenarios. A spatio-temporal grid $\mathcal{R} \in \mathbb{R}^3$ can be defined by an ordinary 3D convolution specified by a certain receptive field size and dilation. For each location $\mathbf{q}_0 \in (t, h, w)$ on the output feature map $\mathbf{y}$, the response is calculated by sampling on irregular locations offset by $\Delta \mathbf{q}_n$.

$$\mathbf{y}(\mathbf{q}_0) = \sum_{\mathbf{q}_n \in \mathcal{R}} \mathbf{w}(\mathbf{q}_n) \cdot \mathbf{x}(\mathbf{q}_0 + \mathbf{q}_n + \Delta \mathbf{q}_n) \tag{4}$$

The trajectory convolution can then be viewed as a special case of 3D deformable convolution where the offset map is from the trajectories. Here, the grid $\mathcal{R} = \{(-1, 0, 0), (0, 0, 0), (1, 0, 0)\}$ is defined by a $3 \times 1 \times 1$ kernel with dilation 1. The temporal component of the offset is always 0, *i.e.* $\Delta \mathbf{q}_n = (0, \Delta \mathbf{p}_n)$. The discussion above reveals the relationship with deformable convolution. Therefore, the trajectory convolution can be efficiently implemented similar to the way discussed in [2].

## 3.3 Combining Motion and Appearance Features

The trajectory convolution helps the network to aggregate appearance features along motion path, alleviating the motion artifact by trajectory alignment. However, the motion information itself is important for action recognition. Inspired by the trajectory descriptor proposed in [37], we describe local motion patterns at each position $\mathbf{p}$ using the sequence of trajectory information in the form of coordinates of sampling offsets $\{\Delta \mathbf{p}_\tau : \tau \in [-\Delta t, \Delta t]\}$. This is equivalent to stacking the offset map for trajectory convolution and the original appearance feature map. The offset map is normalized through Batch-Normalization [12] before concatenation. As a result, we achieve the combination of appearance feature and motion information in terms of trajectory with minimal increase of network parameters. Compared with the canonical two-stream approaches, which are based on late fusion of two networks, our approach leads to a unified network architecture and is much more parameter and computation efficient.

## 3.4 The TrajectoryNet Architecture

Based on the concept of trajectory convolution, we design a unified architecture that can align appearance and motion features along the motion trajectories. We call it *TrajectoryNet* by integrating trajectory convolution into the Separable-3D ResNet18 architecture [9, 36]. The 1D temporal convolution component of a (2+1)D-convolutional block is replaced by a trajectory convolution with down-sampled motion field, such as a pre-computed optical flow, in the middle level of the network. The appearance feature map for trajectory convolution is optionally concatenated with the down-sampled motion field to introduce extra motion information. Adding trajectory convolution at higher levels is likely to provide less motion information since spatial resolution is reduced and down-sampled optical flow may be inaccurate. Adding trajectory convolution at lower levels increases the precision of motion estimation, but the receptive field for sampling position is limited.

## 3.5 Learning Trajectory

As discussed in the previous subsection, the trajectory convolution can be viewed as deformable convolution with a special deformation map, that is the motion trajectory in the video. It is capable of accumulating gradients from higher layers via back-propagation. Therefore, if the trajectory can be estimated by a parametric model, we can learn the model parameters using back-propagation as well. The most straight forward approach for this cause is applying a small 3D CNN to estimate trajectories as an mimic of the 2D CNN used in the deformable convolution networks [2]. Preliminary experiments show that this is not very effective. It can be observed that the offsets obtained simply by applying a 3D convolutional layer over the same input feature map are highly correlated to the appearance. On the contrary, the motion representation, which we use trajectory as a medium, has long been considered to be invariant to appearance intuitively and empirically [17, 28]. Therefore, we cannot naïvely adopt the way of learning offsets in [2]. This also reveals the difference between the original deformable convolution for object detection and our trajectory convolution for action recognition: The original deformable convolution attempts to learn deformation of spatial configuration within an

single image while our trajectory convolution tries to model the variation of appearance deformation across neighboring images, despite sharing the similar mathematical formulation.

To tackle such issue, we train another network to predict the trajectory individually as an alternative. In particular, we use MotionNet [51] as the basis due to its lightweightness. It accepts a stack of $(M + 1)$ images as a $3(M + 1)$-channel input and predicts a series of $M$ motion field maps as a $2M$-channel output. Following a "downsample-upsample" design like FlowNet-SD [11], motion fields with multiple spatial resolutions are predicted. The network is trained without external supervision such as ground-truth optical flow. An unsupervised loss $\mathcal{L}_{\text{unsup}}$ [51] is designed to enforce pair-wise reconstruction and similarity, with motion smoothness as a regularization.

Once pre-trained, the MotionNet can be plugged into the TrajectoryNet architecture to substitute the input of pre-computed optical flow. We modify the original model in [51] to produce optical flow map of the same resolution of feature maps where the trajectory convolution operates on. The MotionNet can also be fine-tuned with the classification network. In this case, the loss for network training is a weighted sum of the unsupervised loss $\mathcal{L}_{\text{unsup}}$ and the cross-entropy loss for classification $\mathcal{L}_{\text{cls}}$, written as $\mathcal{L} = \gamma \mathcal{L}_{\text{unsup}} + \mathcal{L}_{\text{cls}}$.

## 4 Experiments

To evaluate the effectiveness of our TrajectoryNet, we conduct experiments on two benchmark datasets for action recognition: Something-Something V1 [8] and Kinetics [19]. Visualization of intermediate features for both appearance and trajectory is also provided.

### 4.1 Dataset descriptions

**Something-Something V1** [8] is a large-scale crowd-sourced video dataset on human-object interaction. It contains 108,499 video clips in 174 classes. The dataset is split into train, validation and test subset in the ratio of around 8:1:1. The top-1 and top-5 accuracy is reported.

**Kinetics** [19] is a large-scale video dataset on human-centric activities sourced from YouTube. We use the version released in 2017, covering 400 human action classes. Due to the inaccessibility of some videos on YouTube, our version contains 240436, 19796 and 38685 clips in the training, validation and test subset, respectively. The recognition performance is measured by the average of top-1 and top-5 accuracy.

### 4.2 Experimental Setups

**Network configuration** We use the Separable-3D ResNet-18 [9] as the base model, if not specified. Starting from the base ResNet-18 model, A 1-D temporal convolution module with temporal kernel size of 3, followed by Batch Normalization [12] and ReLU non-linearity is inserted after every 2-D spatial convolution module. A dropout of 0.2 is used between the global pooling and the last $C$-dimensional ($C$ equals the total number of classes) fully-connected layer.

**Generating trajectories** As stated above, we study two methods to generate trajectories: one is based on variational methods and the other is based on CNNs. For the former, we adopt the TV-L1 algorithm [48] which is implemented in OpenCV with CUDA. To match the size of input feature, two types of pooling are used to down-sample the optical flow field: average pooling and max pooling. For the latter, the MotionNet is trained by randomly sampling images pairs from UCF-101 [30]. The training policy follows the practices in [51].

**Training** The network is trained with stochastic gradient descent with momentum set to 0.9. The weights for 2D spatial convolution are initialized with the 2D ResNet pre-trained on ImageNet [27]. The length of each input clip is 16 and the sampling step varies from 1 to 2. For Something-Something V1, the batch size is set to 64 while for Kinetics, the batch size is 128. On Kinetics, the network is trained from an initial learning rate of 0.01 and is reduced by $\frac{1}{10}$ every 40 epochs. The whole training procedure takes 100 epochs. For Something-Something V1, the epoch number is halved because the duration of its videos is shorter.

**Testing** At test time, we follow the common practice by sampling a fixed number of $N$ snippets ($N = 7$ for Something-Something V1 and $N = 25$ for Kinetics) with an equal temporal interval. By

cropping and flipping four corners and the center of each frame within a snippet, 10 inputs are obtain for each snippet. The final class scores are calculated by averaging the scores across all $10N$ inputs.

## 4.3 Ablation Studies

**Trajectory convolution**    We first evaluate the effect of using trajectory convolution in the Separable-3D ResNet architecture in Table 2. Consistent improvement of accuracy can be observed if trajectory convolution is used. Then, we study the effect of incorporating trajectory in different locations. Adding trajectory convolution increases the top-5 accuracy but the top-1 accuracy saturates. In the remaining experiments, we use only 1 trajectory convolution at the `res3b1.conv1` block, if not specified.

Since we did not see remarkable gain, we conjecture that this is because the used trajectory is derived from the optical flow down-sampled via average pooling. The optical flow is already smoothed with TV-L1 and the extra average pooling degrades the quality more. To verify this, we preform an additional experiment by replacing average pooling with max pooling. This alternative down-sampling strategy preserves more details without degrading the trajectory. Furthermore, as will be shown in Table 4, using trajectory learned from MotionNet leads to higher accuracy. This indicates that the performance of TrajectoryNet highly depends on the quality of trajectory.

Table 2: Results of using trajectory convolution in different convolutional layers in the Separable-3D ResNet-18 network. The accuracy is reported on the validation subset of Something-Something V1.

| Usage of Traj. Conv. | Down-sample Method | Top-1 Acc. | Top-5 Acc. |
|---|---|---|---|
| None | None | 34.30 | 65.66 |
| res2b1.conv1 | Avg. Pool | 34.49 | 66.23 |
| res3a.conv1 | Avg. Pool | 34.79 | 66.21 |
| res3b1.conv1 | Avg. Pool | 34.96 | 66.24 |
| res3b1.conv1,2 | Avg. Pool | 34.72 | 66.89 |
| res3b1.conv1 | Max Pool | 36.04 | 67.72 |

**Combining motion and appearance features**    We compare the results of incorporating motion information into the trajectory convolution in Table 3. We can clearly see the improvement of more than 1% after encoding a 4-dimensional feature map of trajectory coordinates. We compare with several other methods, such as the early spatial fusion by concatenation with motion feature map [7] and the late fusion used in the two-stream network [29]. Though there is still an apparent gap between ours and the late-fusion strategy, our fusion strategy achieves notable increase with negligible increase of parameters. And it also completely removes the computation for running a motion-stream recognition network.

Table 3: Results of incorporating different sources of input into the trajectory convolution in the Separable-3D ResNet-18 network. The *ft.* denotes the feature map. The accuracy is reported on the validation subset of Something-Something V1.

| Source | Usage of Traj. Conv. | # param. | Top-1 Acc. | Top-5 Acc. |
|---|---|---|---|---|
| appearance | res3b1.conv1 | 15.2M | 34.96 | 66.24 |
| appearance + motion (ft.) | res3b1.conv1 | 15.9M | 35.24 | 67.22 |
| appearance + trajectory (# dim=4) | res3b1.conv1 | 15.2M | 36.08 | 67.72 |
| two-stream S3D (late fusion) | None | 30.4M | 40.67 | 72.79 |

**Learning trajectory**    Here we compare the learned trajectory against pre-computed optical flow from TV-L1 [48]. We choose two architectures of MotionNet: one accepts one image pair and outputs one motion field (denoted by `MotionNet-(2)`), and the other accepts 17 consecutive images and produces 16 motion fields (denoted by `MotionNet-(17)`). We study three training policies: (1) fixing the MotionNet once it is pre-trained; (2) fine-tuning the MotionNet with the classification cross-entropy loss; and (3) fine-tuning the MotionNet with both the unsupervised loss and classification loss. The loss weight $\gamma$ is set to 0.01. The results are listed in Table 4. It turns out that the trajectories learned by both MotionNet-(2) and MotionNet-(17) outperform those derived from TV-L1 [48]. It is interesting to observe that jointly training MotionNet and TrajectoryNet will yield lower accuracies than freezing MotionNet unless the unsupervised loss is introduced. We conjecture that the existence

of $\mathcal{L}_{\text{unsup}}$ can help to maintain the quality of trajectories by enforcing the pair-wise consistency. The necessity of multi-task fine-tuning may also explain the difficulty of using shallow convolutional modules with random initialization to estimate the trajectory, which we have discussed in Sec 3.5.

Table 4: Results of learning trajectory. The settings are elaborated in the body part.

| source of trajectory | fine-tune weight | unsup. loss | Top-1 Acc. | Top-5 Acc. |
|---|---|---|---|---|
| TV-L1 | - | - | 34.96 | 66.24 |
| MotionNet-(2) | ✗ | ✗ | 36.37 | 67.74 |
| MotionNet-(2) | ✓ | ✗ | 34.72 | 65.59 |
| MotionNet-(2) | ✓ | ✓ | 36.91 | 68.47 |
| MotionNet-(17) | ✗ | ✗ | 35.69 | 66.82 |
| MotionNet-(17) | ✓ | ✗ | 35.25 | 66.65 |
| MotionNet-(17) | ✓ | ✓ | 36.69 | 68.52 |

**Trajectories with step greater than one**    Here we evaluate the model which accepts an input of 16 frames but at a sampling step of 2. To be more specific, we collect a consecutive of 32 frames and randomly sample one frame for every two neighboring frames. This enlarges the effective coverage of the architecture, *i.e.* from 16 to 32, while keeping the computation the same. With the strategy of learning trajectory mentioned above, the TrajectoryNet can still improve over the baseline. This also reflects the flexibility of learnable trajectory, since pre-computed optical flow has to be re-run for the whole training set under such circumstances.

Table 5: Results of using trajectories with step greater than one.

| # of frame × step | Effective coverage | Usage of Trajectories | Top-1 Acc. | Top-5 Acc. |
|---|---|---|---|---|
| 16 × 2 | 32 | None | 42.47 | 74.57 |
| 16 × 2 | 32 | MotionNet-(17)-ft.-unsup. | 43.32 | 74.85 |

**Runtime Cost**    In Table 6, we report the runtime of the proposed TrajectoryNet with two settings: (1) the one whose trajectories are from pre-computed TV-L1 (time not included) and (2) the one whose trajectories are inferred from MotionNet-(17) (time included). Compared with its plain counterpart, the TrajectortNet with pre-computed TV-L1 incurs less than $10\%$ additional computation for the operation of trajectory convolution. It takes TrajectoryNet with MotionNet-(17) an extra 0.137 second for network forward compared to TrajectoryNet with TV-L1, which can be ascribed to the forward time of the MotionNet plugged in.

Table 6: Runtime comparison of TrajectoryNet and the counterpart. The network is tested on a workstation with Intel(R) Xeon(R) CPU (E5-2640 v3 @2.60GHz) and Nvidia Titan X GPU.

| Method | Net. forward (sec) | $\Delta t$ (sec) |
|---|---|---|
| S3D | 0.390 | - |
| TrajectoryNet (TV-L1) | 0.426 | +0.036 |
| TrajectoryNet (MotionNet-(17)) | 0.563 | +0.137 |

## 4.4 Comparison with State-of-the-Arts

We compare the performance of our TrajectoryNet with other state-of-the-art methods. The results on Something-Something V1 [8] and Kinetics [19] are shown in Table 7 and Table 8 respectively. For Something-Something V1, we use 16 frames with a step of 2 as input and apply MotionNet-(17) to produce trajectory. Motion information encoded by trajectory is used optionally. On Table 7, we can see that our TrajectoryNet achieves competitive results with state-of-the-art models including those with deeper models or those pre-trained on larger models. After pre-training on Kinetics, the accuracy is boosted to a new level. For Kinetics, a MotionNet-(2) is used. On Table 8, the TrajectoryNet improves the Separable-3D baseline. With 16 input frames at a step of 2, it performs on par with models with similar model complexity.

## 4.5 Visualization

We present a qualitative study by visualizing the intermediate feature of our TrajectoryNet in Figure 2. Given a pair of two consecutive images on top of the first column, we first compare the feature map at

Table 7: Comparison with state-of-the-art methods on the validation and test set of Something-Something V1. The performance is measured by the Top-1 accuracy.

| Method | Backbone network | Pre-train | Val Top-1 |
|---|---|---|---|
| 3D-CNN [8] | C3D | Sports-1M | 11.5 |
| MultiScale TRN [50] | BN-Inception | ImageNet | 34.4 |
| ECO lite [52] | BN-Inception + 3D-ResNet18 | Kinetics | 46.4 |
| Non-local I3D + GCN [44] | ResNet-50 | Kinetics | 46.1 |
| TrajectoryNet-MotionNet-(17) w/o. motion | ResNet-18 | ImageNet | 43.3 |
| TrajectoryNet-MotionNet-(17) w/. motion | ResNet-18 | ImageNet | 44.0 |
| TrajectoryNet-MotionNet-(17) w/o. motion | ResNet-18 | Kinetics | 47.8 |

Table 8: Comparison with state-of-the-art methods on the validation subset of Kinetics. The performance is measured by the average of Top-1 and Top-5 accuracy.

| Method | Backbone network | Pre-train | Val. Avg. Acc. |
|---|---|---|---|
| TSN (RGB) [42] | BN-Inception-v2 | ImageNet | 77.8 |
| I3D (RGB) [1] | BN-Inception-v1 | ImageNet | 81.2 |
| Nonlocal-I3D (RGB) [43] | ResNet-101 | ImageNet | 85.5 |
| R(2+1)D (RGB) [36] | ResNet-34 | Sports-1M | 82.6 |
| C3D [35] | ResNet-18 | - | 75.7 |
| ARTNet w/. TSN [40] | ResNet-18 | - | 80.0 |
| Separable-3D (RGB, $16 \times 1$ frames) | ResNet-18 | ImageNet | 76.9 |
| TrajectoryNet-MotionNet-(2) ($16 \times 1$ frames) | ResNet-18 | ImageNet | 77.8 |
| TrajectoryNet-MotionNet-(2) ($16 \times 2$ frames) | ResNet-18 | ImageNet | 79.8 |

the layer of `res3b1.conv1`, *i.e.* on which the trajectory convolution is applied, at the bottom of the first column. We can observe a visible spatial shift between the two images' high response regions, which conforms to our assumption that feature map is not well aligned due to object movement. We also demonstrate different types of trajectories that we use in the experiments, namely the TV-L1 [48] optical flow and prediction from MotionNet before and after finetune on the second, third and fourth column. We can see that the motion estimation by the original MotionNet is less smooth than TV-L1 especially in the background regions. For foreground objects, however, MotionNet does well and can sometimes produce motion with more rigid shape, *e.g.* the hand on the left example of Figure 2. Also, the joint training further improves the quality of trajectories.

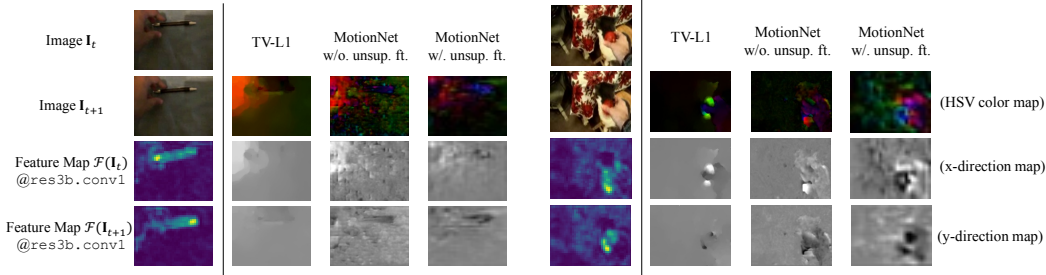

Figure 2: Visualization of the intermediate feature of the TrajectoryNet. These two image pairs depict the action of "moving something (a pen) down" and "trying but failing to attach something (a ball) to something (a cat) because it doesn't stick." For each block, the first column show a pair of input images and their corresponding feature map at the layer of `res3b1.conv1`; the second, third and fourth column show the optical flow field generated by TV-L1 algorithm and learned by MotionNet before and after finetuning (The motion field encoded in HSV color map as well as the components of x-axis and y-axis are shown from top to bottom). The figure is best viewed in color.

## 5   Conclusion

In this paper, we propose a unified end-to-end architecture called TrajectoryNet for action recognition. The approach is to incorporate the repeatedly proven idea of trajectory modeling into the Separable-3D network by introducing a new operation named trajectory convolution. The TrajectoryNet further combines appearance and motion information in a unified model architecture. The proposed architecture achieves notable improvements over the Separable-3D baseline, providing a new perspective of explicitly considering motion dynamics in the deep networks.

**Acknowledgment** This work is partially supported by the Big Data Collaboration Research grant from SenseTime Group (CUHK Agreement No. TS1610626), and the Early Career Scheme (ECS) of Hong Kong (No. 24204215).

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
