[Reviews · NeurIPS 2018]

Reviewer 1



UPDATE: Thank you to the authors for addressing my concerns. With the new version of Table 1, and the clarification of ResNet-18 vs BN-Inception, my concern about the experimentation has been addressed -- there does seem to be a clear improvement over classical 3D convolution. I have adjusted my score upwards, accordingly. === Recently, a number of new neural network models for action recognition in video have been introduced that employ 3d (spacetime) convolutions to show significant gains on large benchmark datasets. When there is significant human or camera motion, convolutions through time at a fixed (x,y) image coordinate seem suboptimal since the person/object is almost certainly at a different position in subsequent frames. This paper proposes a solution, “trajectory convolution”, which performs temporal convolution along the path traced out by optical flow -- in essence following the moving object. Think of this as motion-stabilized 3D convolution. Further, the motion trajectory can be included in the feature representation, providing an elegant way to combine both appearance and motion information in a single network. The idea is straightforward, and the paper is clearly written, but the experimental evidence is not enough to justify the claim that trajectory convolution is superior to classical space-time convolution. Quality: My primary concern with this paper is in the experimental evaluation. Specifically in Table 6, Separable-3D, presumably [45] the number reported is 76.9. But the source paper https://arxiv.org/pdf/1712.04851.pdf (see Table 6) reports top-1 & top-5 accuracies of 77.2 and 93.0 respectively, giving an average accuracy of 85.1. Even using the RGB-only stream (see table 4) the numbers are 74.84 & 91.93, giving an average 83.4. The other comparable space-time convolution method, R(2+1)D, achieves 82.6. The TrajectoryNet performance of 79.8 does not support the claim that trajectory convolution leads to superior performance over classical (separable) convolution. Table 5, with performance on Something-Something, would be more convincing if it included R(2+1)D and S3D, as well as fewer blank values. Further, the differences between results is very small in several tables, especially Table 1. Would it be possible to include error estimates? That would help to gauge which differences are significant. Clarity: The explanation and writing is very good. Just a few minor points of clarification: In general, what is the additional runtime cost of doing trajectory convolution vs. rectangular convolution? In section 3.3, by how much does the number of parameters actually increase? In Table 2 & corresponding text, please be a little more clear about how exactly the “4-dimensional appearance map of trajectory coordinates” is formed. In table 6, Separable-3D should cite [45] if I’m not mistaken. Originality: There has been much recent work exploring exactly how best to combine appearance and motion (optical flow) in action recognition networks. The specific approach proposed is a new combination of familiar techniques. The related work section is comprehensive and well-written. Significance: This paper would be more significant with experimental results that clearly validate the claim that trajectory convolution is superior to classical 3D convolution.

Reviewer 2



This work proposes a novel convolutional operation and corresponding network architecture for action recognition in videos. Starting from 3D convolutions, the authors introduce a trajectory convolution approach where the convolutional aggregation is performed in a dynamic manner through tracked points, rather than fixed locations. Experiments and ablation studies show the effect of the primary components of the new network, as well as decent overall performance on two large-scale action datasets. Paper strengths: - The first strength of the paper is that it is well written. The main motivation of the paper is presented in a clear and intuitive manner. Where current 3D convolutional operators perform an aggregation within a fixed cube, this work posits that the temporal aggregation should be dynamic by following points over time. This way, it should be possible to aggregate features based on the primary motion of interest. Furthermore, from a presentation point of view, the embedding of the proposed trajectory convolution in recent convolutional literature adds to the clarity and insight of the trajectory convolution itself. The embeddings is also unclear at several instance, as discussed later. Positive embedding discussions include: -- The difference and novelty with respect to the Trajectory-Pooled Deep-Convolutional Descriptor (TDD) is clear and fair. -- A link is made with deformable convolutions, which helps the authors with the implementation of the trajectory convolution. -- A link is also made with separable convolutions. This makes sense in isolation, although it is unclear how it all combines with deformable convolutions. - The second strength of the paper is that care is taken to perform the necessary ablation studies. The trajectory convolution is evaluated, as well as the fusion with motion and the learning trajectory component. This paints a fair picture of the method and where it can best be employed. Such ablation studies are a necessity to gain insight into the method and to understand how it works. Paper weaknesses: - The first weakness of the paper is that the paper is unclear in its method and missing details. -- While Section 3.1 is clear, the rest of Section 3 is unclear at several occasions. In Section 3.2, a link is made with deformable convolutions with the note that the trajectory convolution can be implemented in this manner (lines 152-153). Section 3.4 in turn discusses the implementation by extending separable 3D-convolutional networks. Does this mean that the proposed trajectory convolution combines both notions? If so, these subsections should be combined and rewritten to better describe how the proposed approach incorporates both. If not, the distinction should be made more clear. -- Section 3.3 discusses the addition of motion information by adding offset coordinates as features. This is also done e.g. in Dense Trajectories themselves and in TDD. From Section 3.3, it is however unclear how this is performed in this context. To what are these offsets added and where in the network are they used? The explanation on lines 159-160 is not intuitive. A figure would also really help here. -- Section 3.5 is not clear at all. The method jumps from using a small 3D CNN to estimate estimate trajectories, to preliminary experiments, to MotionNet, to FlowNet-SD, to replacing optical flow. The same subsection discusses multiple topics in half a page without proper explanation and details, making it unnecessarily hard to read. Please rewrite this subsection and again consider a figure to make things clear to the reader. -- On a general note, it remains unclear from the paper which elements form a trajectory. I.e. which points are tracked, how densely this is done, etc. Without these details, the paper remains abstract and hard to interpret. - The second weakness of the paper is that the experiments are not fully convincing in three aspects. -- The first aspect is that the proposed trajectory convolution does not add much to the current 3D convolutional network setup. While large improvements are not always a requirement, it is difficult to assess whether the improvements in Tables 1 through 3 are even due to the method, or due to randomness within the networks. Rerunning the baselines and methods multiple times with different seeds followed by significance tests seem like a requirement for these ablation studies given the small improvements. On face value, the current improvements are small. -- The second aspect is that it remains unknown why the method works and when it fails. Next to the missing details on the trajectories themselves, their impact also remains abstract and unknown. Figure 2 provides a qualitative result, but does not really add to the understanding of the trajectory convolution itself. What do the trajectories focus on? Where and when are the trajectories used within a video? For which actions do they work and why? For which actions to they not work? How different are the learned trajectories from straight lines through the temporal axis? The lack of such analyses limit the impact and insight into the proposed network. The listed strengths and weaknesses of the paper paint an overall interesting work. The trajectory convolution is a natural extension from the current literature on action recognition. Furthermore, the paper is clearly written. The paper does however have several weaknesses that require attention and warrant discussion. The method itself is not well explained and requires thorough rewriting, while the experiments are not fully convincing due to lack of effect. It should however be noted that the method itself makes sense within the current action recognition literature and the lack of big improvements should not be a direct cause for rejection, as it might still be an interesting result for researchers investigating similar avenues.

Reviewer 3



In this paper, the authors propose a TrajectoryNet for action recognition in videos. With the guidance of optical flow, a trajectory convolution operation is designed to learn the dynamics along the motion paths. *Positive: The idea is interesting. In practice, the feature maps across frames are not well aligned. Hence, Learning temporal dynamics along the motion paths seems to be a preferable design, compared to the standard temporal convolution. *Negative: (1) Model Design # The trajectory is generated from optical flow. This would require either the high computation (e.g., TV-L1) or extra model training (e.g., MotionNet). Both cases reduce the conciseness and practicability of 3D CNN for action recognition in videos. # Since the approach requires optical flow anyway, have you tried trajectory convolution on optical-flow-stream 3D CNN? (2) Experiments # In Table 2, how to generate motion trajectories? TV-L1 or MotionNet? When using TV-L1, the pre-computation burden of optical flow is ignored. When using MotionNet, the parameters of this model is ignored ([48]: Tiny-Motionnet / Motionnet is 8M / 170M). Furthermore, two-stream S3D achieved the best result with a large margin. In this case, it is controversial to exchange high accuracy for model complexity, especially under the same condition (i.e., both two-stream S3D and TrajectoryNet requires optical flow). # In Table 1 and Table 6, the influence of trajectory convolution seems to be small. Especially in Table 6, most of state-of-the-art approaches do not require the guidance of optical flow.